



# Abutting faults: a case study of the evolution of strain at Courthouse branch point, Moab Fault, Utah

Heijn van Gent[1,2], Janos L. Urai[1]

[1]Structural Geology, Tectonics and Geomechanics, RWTH Aachen University, Lochnerstrasse 4-20, Aachen, Germany

[2]Now at: Shell International Exploration and Production, The Hague, the Netherlands

*Correspondence to*: Heijn van Gent (Heijnvangent@hotmail.com)

**Abstract.** Slip planes and slip directions of subsequent generations of faults were measured in the interaction damage zone two abutting faults in porous sandstones, to understand the evolution of paleostress/paleostrain evolution.

The Courthouse branch point, of the Moab Fault in SE Utah, is a much-studied spectacular outcrop of two abutting faults. It
shows a wide range of deformation structures and fault related diagenesis: striated slip planes, deformation bands, veins, Liesegang bands and copper-rich mineralization.

By combining our own measurements with published results on the relative age of these structures, we classified our data in four sets. Using the Numeric Dynamic Analysis (NDA) to calculate the orientation of the kinematic axes we found three different paleo-extension directions in the four sets, recording the evolution of stress/strain axes during the abutting process.
The first phase of deformation direction is the regional extension in the NE-SE direction. As the second fault approached the main fault from its footwall side and the two faults start to become kinematically linked, the extension direction changed so that the overall extension became perpendicular to the approaching fault (NW-SE). Finally, the extension direction changed back to perpendicular to the first segment (NE-SW), when the two faults become geometrically-linked and regional extension dominates..

## 1 Introduction

Although at the regional scale the stress field and kinematics of extensional tectonics is well established Anderson (1951), less is known of the interaction of faults at the local scale (Walsh and Watterson, 1991; Peacock, 2002; Nicol et al., 2010; Duffy et al., 2015; Delogkos et al., 2017; Peacock et al. , 2017a), although this is where damage zones and the associated flow of geofluids occur (see for example Walsh et al., 1998; Lonergan, 2007; Rotevatn et al., 2009a; Yale, 2003; Eichhubl et al., 2009;
Duffy, 2015. Peacock et al. (2017a, b) suggested that flow is already impacted in the "interaction damage zone" ("the area of deformation caused by the interaction between two or more faults") as the faults are approaching each other before becoming kinematically linked (Peacock et al., 2017a). The partitioning of strain at this scale can compartmentalize reservoirs or provide conduits for fluid flow depending on the mode of failure (Aydin and Johnson, 1978; Johansen et al., 2005; Fossen and Bale, 2007; Rotevatn et al., 2007; Bozkurt Çiftçi and Bozkurt, 2007; Urai et al., 2008; Eichhubl et al., 2009; Ferrill et al., 2009;
Rotevatn et al., 2009b; van Gent et al., 2010a, Vrolijk et al., 2016, Kettermann and Urai, 2015).



Minor structures in damage zones can extend several tens to several hundreds of meters away from the fault core and their evolution in porous sandstones has been the subject of numerous studies (Antonellini and Aydin, 1994; Aydin and Johnson, 1978; Cartwright et al., 1995; Cowie and Shipton, 1998; Shipton and Cowie, 2001; Shipton and Evans, 2002; Shipton and Cowie, 2003; Davatzes and Aydin, 2003; Davatzes et al., 2003, 2005; Crider and Peacock, 2004; Kim et al., 2004; Johansen

et al., 2005; Fossen and Bale, 2007; Fossen, 2010; Soliva et al., 2016; Peacock et al, 2017a, 2017b).

In this study we focus on the changes in the stress and strain field in a single-tip fault interaction (Fossen et al., 2005), where one fault grows towards another. Peacock et al. (2017a) call this an "abutting fault interaction", where the synchronously growing, abutting fault is synthetic to the main fault. When the two faults become hard-linked they move together, without crosscutting. We use the orientation and slip direction of different generations of minor faults, in a large outcrop as input for

paleostress and paleostrain analysis. We analyze changes in the extension direction and related local stress field over time, which we correlate with the transition from isolated faults (stage 1), through interacting, kinematically linked faults (stage 2), to an abutting fault arrangement, where the faults are geometrically linked (stage 3).

## 2 Regional Geology

The study area is located on the Colorado Plateau, near Moab, UT, where evaporites of the Paradox formation were deposited

on basement in the Carboniferous (Doelling, 2002; Nuccio and Condon, 1996). In the Jurassic, major salt-cored anticlines (Fig. 1a) developed in the Moab area (Doelling, 2002; Foxford et al., 1996), with numerous salt-related faults and salt welds in the overburden. The Moab salt related fault is described in detail in Foxford et al. (1998), and many other publications. Maximum burial depths of 2-3 km and temperatures of around 80 °C were reached in the Entrada sandstone (Nuccio and Condon, 1996 ; Fossen, 2010).

The rocks in the study area (Doelling, 2002; Foxford et al., 1996) range in age from the Jurassic to Cretaceous. The Lower Jurassic aeolian Navajo Sandstone is overlain by the Middle Jurassic Dewey Bridge Member, a calcareous, medium- to fine-grained sandstone. The Slick Rock Member is a part of the Middle Jurassic Entrada Fm. This is a reddish to orange, fine grained and cross-bedded aeolian sandstone, which is overlain by the grayish-yellow sandstones of the Moab Member (Curtis Fm). Directly overlying the Moab Member is the terrestrial Morrison formation (late Jurassic), consisting of siltstones, fine

grained sandstones, nodular limestones and conglomerates, also containing dinosaur bones.

The Moab fault (Fig. 1) is a salt related normal fault with a maximum throw of ~950 m near the entrance of the Arches National Park and a length of around 45 km (Foxford et al., 1996, 1998). The fault offsets rocks from the Late Carboniferous to the Cretaceous, and was active from the Triassic to mid-Jurassic, and again from mid-Cretaceous to Early Tertiary, when the salt dissolved and the salt structures collapsed (Foxford et al 1996, 1998), although further tectonic activity cannot be ruled out

(Gutiérrez, 2004).

In aerial and satellite photos the exposed sandstones of the Colorado Plateau display a dense network of sub-vertical joints in many locations which formed due to the uplift of the Colorado Plateau in the Tertiary (Cruikshank and Aydin, 1995; Foxford





et al., 1996; Kettermann et al., 2015). Similar structures are visible in the footwall of the Moab Fault in the Moab Member but not within the CHJ and postdate the activity of the Moab Fault (Cruikshank and Aydin, 1995; Fossen, 2010).

## 2.1 Outcrop description

In the study area (Fig. 1b) the Moab fault forms a number of hard-linked strands (Fig 1, Foxford et al., 1996), which likely developed as breached relay ramps (Foxford et al., 1998, Fossen and Rotevatn, 2016). This implies that the main Moab Fault strand (segment A, Figure 1b and 2) existed prior to the approaching fault (segment B) that abuts it. This is opposed to a secondary fault splaying from the main fault. A third segment, (segment C) is exposed further to the west. At the intersection of segment A and B, a spectacular outcrop of abutting faults (Courthouse Junction (CHJ) branch point, Fossen et al., 2005) is found (Fig. 1, Fig. 2 and Fig. 3) (Foxford et al., 1996; Davatzes and Aydin, 2003; Davatzes et al., 2005; Eichhubl et al., 2009; Fossen et al., 2005; Johansen et al., 2005). The two faults abut at an angle of 64°, and the intersection damage zones are exposed in the footwall of Segment A and the hanging wall of Segment B (see box in Figs. 1b and 2a). The typical 'forking geometry' (Simón, 2019) of the only strand of thin deformation bands with more than 5m of slip, (Eichhubl et al. 2009), supports the interpretation that Segment A was abutted by Segment B.

Peacock et al. (2017a) describes the interaction of abutting faults as a continuous process in the steady regional stress field, with three evolutionary stages: (i) the faults are far from each other and do not interact. (ii) the faults interact as they approach each other and become kinematically linked. Interaction can be indicated by deformation structures in the acute bisector. In the final stage (iii) segment B abuts segment A, and they become geometrically linked. At this stage a branch or intersection line will develop, both faults slip parallel to this line, and the faults become hard-linked. The fault rotation and linkage is the result of local stress field perturbation as segment A and B start to interact.

The triangular CHJ outcrop exposes gray-yellow sandstones of the 28 m thick Moab Member (with small outcrops of Morrison Fm on top), while in the Courthouse Canyon to the south older rocks, including the Navajo sandstone are exposed. The Moab Member typically has a porosity of around 20-25% (Antonellini and Aydin, 1994) which is strongly reduced, between 17 and 1% at the CHJ (Eichhubl et al., 2009; Johansen et al., 2005). Foxford et al (1998) constrains the throw along segment A to about of 370 m just south of Courthouse Rock, and between 100 and 20 m in Mill Canyon, just north of Courthouse Rock (Figure 1b). Eichhubl et al. (2009) estimates throw on Segment A about 200 m at the CHJ, and on Segment B in the order of 100 m.

Johansen et al (2005) published a detailed map of the outcrop, showing the distribution, location and interaction between the different structure types mapped. These include joints and three types of deformation bands: Thick deformation bands, Thin deformation bands and Thin Deformation Bands with a Slip plane. All these types occur in bundles. While the 'Thick' (~1–1.5 mm wide) deformation bands are dominated by relatively large grainsize (~50–150 μm) and occasional undeformed grains, the Thin deformation bands are truly cataclastic with grainsizes of <5 μm. The thin deformation bands were subdivided into seven separate evolutionary sets (fig. 12 in Johansen et al. 2005). Occasionally, the Thin deformation bands slip along smooth, corrugated surfaces, which are interpreted as slip planes (Fig. 2 b-e). Some authors interpret these as joints and sheared joints



(Davatzes et al., 2005; Eichhubl et al., 2009). Cross-cutting relationships suggest that the thick deformation bands are older than the thin bands (Johansen et al. 2005). Eichhubl et al., 2009 also report joints, but these formed in opening Mode I (and are not sheared. Some are located inside deformation bands, indicating that some of the jointing postdates cataclasis in cases. Davatzes et al. (2005) use a static-elastic calculation of stress at the junction to explain the late formation of these joints, which
is controlled by a combination of changes in remote tectonic stress, burial depth, fluid pressure, and rock properties.

## 3 Methodology

### 3.1 Paleostress and Paleostrain

The reconstruction of either the paleostress or the paleostrain tensor from fault slip data has a long history, but is still not considered a mature field of structural geology (Simon, 2019). Nevertheless, a range of both Paleostress and Paleostrain
reconstructions have added significantly to our toolbox to understand brittle deformation in crust (Angelier, 1985; Angelier et al., 1985; Bles et al., 1989; Kleinspehn et al., 1989; Choi et al., 2001; Vandycke, 2002; Dyksterhuis et al., 2005; Martínez-Martínez, 2006; Bozkurt Çiftçi and Bozkurt, 2007; Balsamo et al., 2008 ; Sippel et al., 2009; van Gent et al., 2009, 2010 , van Gent et al., 2009, 2010b). Many of these studies draw inferences on the first order, regional scale stress fields (Gölke and Coblentz, 1996; Gruenthal and Stromeyer, 1994; Warners-Ruckstuhl et al., 2013).
Understanding the evolution of third order local stress perturbations around a group of interacting faults (Peacock, et al. 2017a), is mostly based on numerical models (see e.g.: Ackermann and Schlische, 1997; Thomas and Pollard, 1993; Gupta and Scholz, 2000; Maerten et al., 2002;  Davatzes et al., 2005; Nikolinakou et al., 2011) . Outcrop scale examples include changing extension directions inside relay ramps (Bozkurt Çiftçi and Bozkurt, 2007), stress analyses based on focal mechanisms in the Alps (Larroque et al., 1987) , Hawaii (Liang and Wyss, 1991; Wyss et al. 1992) and Iceland (Plateaux et al., 2009), paleostress
changes around fault tips (Balsamo et al., 2008) and interaction of core complexes with their associated fault systems (Martínez-Martínez, 2006).

Paleostrain and paleostrain reconstruction algorithms share many assumptions, although they differ strongly in their approach. Only the orientations and the relative sizes of the principal stresses or strains are reconstructed, not the full tensor. This produces (in the case of paleostress using the 'dynamic' fault-slip analysis) a "reduced stress tensor", and for the 'kinematic
paleostress analysis" a reduced strain tensor. In the case of coaxial deformation these can be considered to coincide with the stress axes and the reduced stress ratio  (Sperner et al., 1993; Sperner, 1996; Ilic and Neubauer, 2005), but the differences in infinitesimal strain (earthquake analysis) and finite strain (fault slip data) have not yet been fully addressed. This reduced stress ratio (R) is the relative size of the different stress (or strain) axes, which equals (Sigma1-Sigma3)/(Sigma1-Sigma3). For the reduced strain ratio, Sigma 1-3 are replaced by Lamda 1-3. For a more complete discussion of these issues see Angelier (1994)
, Ramsay and Lisle (2000),  Celerier et al. (2012) and Simon (2019).

The kinematic approach is based on the Mohr-Coulomb Criterion (Coulomb, 1776; Mohr 1990) and assumes a high shear stress to normal stress ratio. An example is the Numeric Dynamic Analysis (NDA) which calculates the orientation of the



kinematic axes and the kinematic reduced ratio. These methods treat all faults as being newly formed. A further requirement is that the contraction and extension axes lie in the plane defined by the slip direction and the fault plane normal. This makes

this method unsuitable for the use with faults with oblique striae and reactivated faults (Sperner, 1996). These methods also require the assumption of an angle of internal friction (generally assumed to be 30° neglecting the natural variability of this parameter (Sperner, 1996; Sperner et al., 1993).

The dynamic method is based on the assumption that slip on a fault plane will be parallel to the direction of maximum resolved shear stress on the fault plane (Wallace, 1951; Bott, 1959). The Direct Stress Inversion Method (DSI) (Angelier, 1979; 1984;

1990) is an example of a Wallace and Bott type paleostress analysis, where this argument is mathematically inverted to establish the reduced stress tensor using the orientation and slip direction of many faults. Two important assumptions of stress inversion are that the sampled stress field is homogeneous on the scale of the study area, and that each fault is independent of the others, they are not coeval, and they do not interact (Simon, 2019)

Papers by Celerier et al., (2012) and Simon (2019) discuss the similarities between paleostress and paleostrain, and the simple

geometric relationships that are sometimes observed between strain and stress axes when the fault and slip orientations are clusters. However, in cases when these orientations are more scattered, the two approaches yield different results. Celerier et al. (2012) argue that 'by summing each fault contribution to obtain global strain tensor, strain tensor may be well suited for averaging strain within a volume'. The inverse methods of stress analysis on the other hand may be more suited to 'separate heterogeneous data'..

Separating heterogeneous fault orientation – slip direction pairs into homogenous subsets which can reasonably be considered to originate from a single phase, is one of the major challenges in paleostress calculations (see for example Sippel et al., 2009 and references therein), and individual faults that are wrongly assigned, or are overprinted by a later stress state can cause the inversion algorithm to fail. Because in the CHJ the structures have evolved over a long history, but in more or less the same regional stress field, and are interacting with each other, overprinting and gradual stress rotations are expected, a paleostrain

analysis is in our opinion more suited to analyse this outcrop. We will note a number of cases where paleostress and paleostrain axes are very similar. We used the implementation of NDA in the reconstruction program Tectonics FP (Ortner et al., 2002, http://www.tectonicsfp.com/). For completeness and discussion, the paleostress results of the DSI method are also included in the figures.

**3.2 Field observations**

Our focus at the CHJ was to measure the striated slip surfaces in the different generations of slip planes with polished surfaces and clear undulations forming striations which are interpreted to represent the slip direction (see: Angelier, 1994; Doblas, 1998, Fig. 2b and Fig. 2c). Macroscopic observations on samples cut perpendicular to the slip plane show an increase of deformation band frequency towards the slip plane, in addition to a bleaching of the matrix colour (Fig. 2d). The slickensides do not react with HCl. Observations in thin sections show an anastomosing network of cataclastic deformation bands with a

significant reduction of grain size (Fig. 2e), in agreement with Aydin and Johnson, (1978) and Fossen et al. (2007). In some



cases, anastomosing fractures are found inside the deformation bands, filled with an amorphous, brown-yellow iron oxide. We infer that these fractures have formed late and might result from unloading or weathering close to the surface. More, highly detailed optical and Broad Ion Beam polished SEM micrographs of samples through deformation bands from this and other outcrops around Moab can be found in Komoroczi (2014).

We collected 207 separate fault orientation – slip direction measurements from the larger CHJ area (see Fig. 3). The bulk of these measurements (n=123) are collected at the CHJ outcrop (Fig 3a). The remaining data comes from both inside the fault zone and from the footwall of Segments B and C (Fig. 3b) with very similar slip planes. Measurements at the CHJ and those on top of the unnamed tower directly west of the Courthouse Rock are in the Moab Member, while those at the base of the Courthouse Rock (Fig. 3b) are in the Slick Rock and Dewey Bridge members. Those inside the fault zone of Segments B and

C are in either Slick Rock or Moab members. The observed slip planes often form conjugate sets.

Where the slip planes have undulations on meter and smaller scale (See Fig. 4), the different slip directions are generally parallel (Fig. 4d). The observation that slip planes typically occur in the core of a deformation band bundle, is illustrated by this example (Fig. 4b and 4c). To better capture the variability of slip direction, and reduce the effect of individual outliers due to local spurious orientations or measurement errors, we measured the fault plane and striation orientation up-to three times

on a single structure, about a meter apart. In agreement with Johansen et al. (2005), we did not observe slip planes in the set consisting of "thick deformation bands", which are associated with Segment A, but the seven sets associated with the 'thin deformation bands' can all be recognized in our data set of fault orientation – slip direction pairs.

Fault orientation – slip direction pairs measured outside of the Branch Point were grouped into four sets (I-IV) from East to West (see Fig. 3b), based on their geographical spread. Location I contains many observations along the base of the Courthouse

Rock in the footwall of Segment B, directly south of the branch point (see fig. 1 & 5).

No polished slip planes and slickensides are observed on Segments A and B, but observations of the meter-scale undulations of the main fault structures show slip on these structures was dip-slip (at least in the final stages of movement; Fig. 3a).

## 4 Results and Discussion

### 4.1 Separating the data at the Courthouse Junction

We separated our 123 fault orientation – slip direction pairs from CHJ into the seven consecutive evolutionary sets of thin deformation band bundle sets, following Johansen et al. (2005). Since Set 1 and Set 2, Set 3 and Set 4, and Set 5, Set 6 and Set 7 are conjugates (the average poles are at an angle of around 60°), they are combined in single sets: "Set 1&2", "Set 3&4", and "Set 5, 6 & 7" (see Fig. 6).

### 4.2 Paleostrain results at the Courthouse Junction

The calculated paleostress/paleostrain results at the CHJ are shown in Figure 6. As discussed above, we focus on strain axes calculated using NDA. Results show extension in the NW-SE direction for subset 1&2 and subset 3&4. For set 1&2 the

extension axis has the direction 334/03 and for set "3&4": 336/10, with R-values of 0.48 and 0.46 respectively. The maximum compressive strain axis is near-vertical for Set 1&2 but is towards the north west on the lower hemisphere stereoplot for Set 3&4. The results for subset "5,6&7" show extension in the NE-SW direction, (orientation 219/03) with an R-value of 0.47.

We note that the DSI method produces paleostress axes for this particular set, which are broadly similar to the paleostrain axes, though the R value is very different.

### 4.3 Results in the rest of the study area

Figure 7 shows the paleostrain/paleostress results outside CHJ. At location I, 37 measurements were collected in the footwall of Segment B (Fig. 5). Based on the distribution of slip directions, we separated the data in two sub-sets (set Ia and Ib). Since

we did not observe cross-cutting relations in the field, we cannot determine age relationships. The resultant stress states for the full data-set at Location I and sub-set Ia and Ib as well as the results of location II and III are shown in Fig. 7. By separating the data from location I into subset Ia and Ib, the spread in the histogram reduces for both methods. This histogram represents the relative frequency of the angular mismatch between the orientation of slip on a single fault plane, and the calculated slip direction on that plane, providing a quality control tool. Distributions on this histogram should be half-bell shaped. The

paleostress results for Subset Ia are broadly similar to the paleostrain results.

In location II and location III, the paleostress and paleostrain axes align. As these locations are in the footwall of Segment B, away from major fault interactions, it seems reasonable that here the main paleostress requirements of a homogeneous stress field and independency between the faults are met. As a result NDA and DSI produce results that are similar.

The data at position IV in Fig. 3 is insufficient to calculate a stress or strain tensor (at least four, but preferably 15 independent

measurements are required, (Sperner, 1996; Sperner et al., 1993, Simon, 2019). The sample location east of the stream in location IV (red great circle in Fig. 3) has an anomalous orientation compared to the north dipping fault at the western location (blue great circles), but slip directions are near parallel, suggesting reactivation of an earlier structure. The slip directions do not fit the expected NS extension. This might be due to the perturbation of the stress field by the nearby branch point on Segment C with the minor fault structure, (highlighted with a* in Fig. 1b.

**5 Interpretation and discussion**

Johansen et al. (2005) discuss the relative timing of the different sets of deformation bands, based on cross-cutting relationships, and our observations are consistent with this interpretation. Sets 1 and 2 are conjugates but no crosscutting relations are observed. Sets 3 and 4 are offset by sets 5 and 6, and set 7 shows no cross-cutting relations. Johansen et al. (2005) conclude that Sets 1 and 2 and sets 3 and 4 can be considered to have developed roughly simultaneously, based on similar

orientations. Our study shows that Set 1&2 and Set 3&4 formed under similar extension directions, further strengthening this argument. Similarly, Sets 5,6&7 are also kinematically consistent, thus we assume them to have formed roughly at the same time. In this set, the paleostress axes are consistent with the paleostrain axes, while this is not the case for combined Sets 1&2





and Sets 3&4. Johannsen et al. (2015) describe Set 5, 6 & 7 to be formed directly after Segment B abuts against Segment A, while Set 1&2 and Set 3&4 formed during the approach and initial interaction of Segments A and B, and probably with a more
complex history with gradually rotating deformation axes. Similarly to the situation described above for locations II and III, Set 5,6&7 is interpreted to be the only set where the requirements of a homogeneous stress field and independency between the faults are met. Thus both methods produce broadly similar results here. The same can be observed for the paleostress results of Location Ia. Perhaps the slip directions of subset Ia are more homogenous, as they formed earlier, and are only influenced by the deformation on Segment B. Once Segment A and B become hard-linked, the interaction between these faults results the
DSI-method to fail. Unfortunately, no crosscutting relations were found to test this interpretation.

## 5.1 Deformation during the transition from kinematically to geometrically linked abutting faults

In this study we used both kinematic and dynamic analysis of fault slip data. The NDA method yields consistent results over the study area, which is a reasonable results if a full mechanical analysis is not within reach (Celerier et al. 2012). In those locations where we can reasonably assume a homogenous stress field and no fault interaction, we found similar stress and
strain axes. In the other locations where the relation between calculated stress and strain axes diverges, we infer that this is because DSI cannot separate the effects of the gradually rotating stress field as Segment B approaches Segment A. Using the analysis of Johannsen et al. (2005) to separate data from different generations, the strain-based NDA method which averages strain in a volume, can analyze this evolution.

 Figure 8 summarizes the evolution of the stress state in three phases, assuming coaxial deformation: (i) as Segment A and B
are  isolated (Fig. 8a), (ii) kinematically linked (Fig. 8b) and (iii) geometrically linked (Fig. 8c).

For phase (i), our observations are consistent with the interpretation of Johansen et al. (2005) that Segment A formed first. Since no striated slip planes are observed here, we assume a vertical $\sigma_1$ and use the bisector to estimate $\sigma_3$ at 040-045°, perpendicular to Segment A. This is interpreted as the regional stress field. The Thick deformation bands also formed during this phase (Johansen et al., 2005)

In phase (ii) (Fig. 8b), the two faults start to interact. Sets 1&2 form ahead of the fault tip of Segment B and in the interaction damage zone, in a heterogeneous, rotating stress field as Segment B approaches Segment A. This rotation eventually causes a second group of deformation structures (Set 3&4) to form. The difference in orientation of the extension axis between the results of Set 1&2 and Set 3&4 is small, but the maximum principle strain (Lambda 1 in Fig. 6) rotates away from vertical.

In Phase (iii) the segments A and B connected and became geometrically, or hard-linked. The stress state in the Courthouse
branch point rotates to NE-SW tension as seen in sets 5,6&7 (Fig. 8c).

The relative timing of the observations outside CHJ is poorly constrained, but extension at locations I, II and III was roughly N-S (Fig. 8b). At location IV our data suggest that extension was with a slight anticlockwise rotation. If we assume that Set Ia formed before Ib, the rotation caused by the linkage of Segment A and B corresponds to the minor rotation of axes at this location. In locations II and III we infer that the stress states were not influenced by the connection of Segment A and B, and
deformation axes are assumed to be relatively stationary here.

The changes in orientation of the principal stresses over time in CHJ are interpreted to show the local, third order stress evolution, in a regional stress field: the Moab fault formed during halokinesis (Foxford et al. 1998) and in a NE-SE regional extension. Thomas and Pollard (1993) show that near-tip stresses in models of en-echelon mode I fractures dominate over the regional stresses, and these stresses change before oblique intersection. Davatzes et al. (2005) model the stress state at a branch

point by combining observations from the Courthouse branch point and the branch point close to location IV (Fig. 3). Their models are iterative and the calculated stress orientations are similar to our results.

Peacock at al. (2017a) discuss how the interaction damage zone can influence fluid flow before the faults become geometrically linked. Eichubl et al. (2009), infer most diagenetic cement deposits in the CHJ to be associated with Set 5,6&7 which formed after the faults became hard-linked. They identify the structures associated with this set as joints and not as deformation bands.

Johansen et al. (2005) however shows that joints in the CHJ formed inside a deformation band (an observation shared by us), and thus jointing post-dates the formation of deformation bands. This suggest that the diagenetic overprint in the CHJ is later then the abutting of the fault.

## 6. Conclusions

We analyzed subsequent generations of striated minor fault planes in the overprinting damage zones of interacting normal

faults in porous sandstone, at Courthhouse Junction, near Moab, UT.

In this case study of a single-tip interaction of abutting faults, we find a near 90° rotation of the extension direction (from NE-SW to NW-SE) as Segment B approaches Sement A to become kinematically linked. Once the faults abut and become geometrically linked, there is a second, large rotation, back to the original, regional extension direction. This evolution represents a third order local change of the regional stress field.

## 275 Acknowledgements

We thank Wintershall GmbH for financial support by the "Tight Gas Initiative" of Wintershall and RWTH Aachen University. We further thank Bert de Wijn and Andreas Frischbutter (Wintershall) as well as Janneke IJmker, Steffen Abe, Frank Strozyk and Myron Thomas and Zoltan Komoroczi for discussions and critical comments to the manuscript.

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





**Figure Captions**

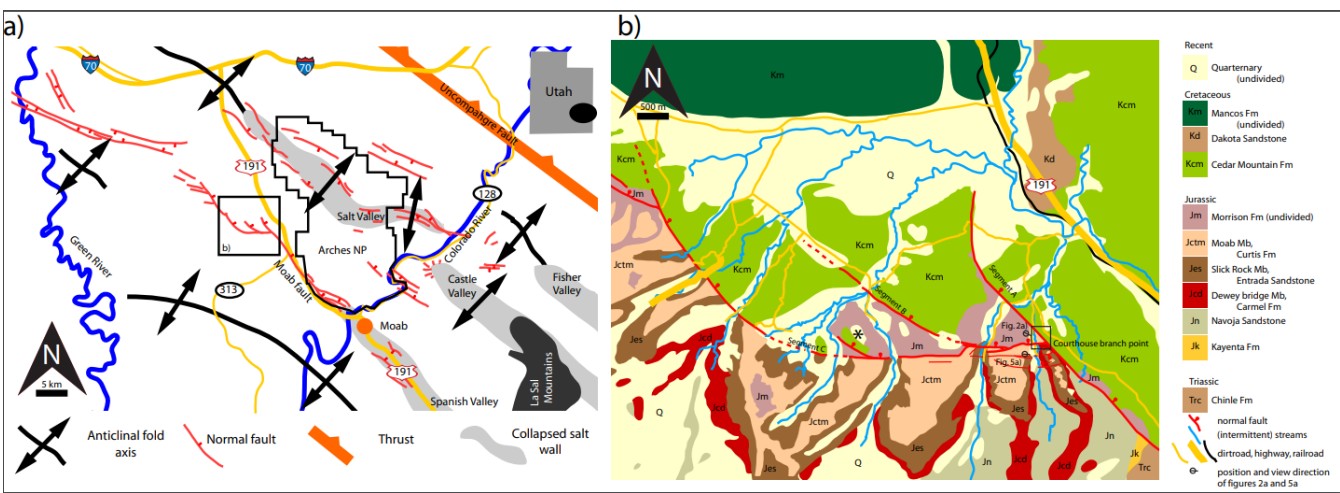

**Fig. 1: a) Simplified geological map of the Moab area. b) Geological map of the study area. The box denotes the position of the Courthouse Junction Branch point, and the locations and view direction for Fig. 2a and Fig. 5 are shown. Maps are modified from: Nuccio and Condon, 1996, Doelling, 2002, Gutiérrez, 2004, Fossen et al., 2005, Johansen et al., 2005, Eichhubl et al., 2009, including our own observations. Coordinates of top left and bottom right corners of the maps (UTM): a) 38.963783, -110.183635; 38.468487, -109.139748, b) 38.754977, -109.810685; 38.691298, -109.710322.**






**Fig. 2: Outcrop and sample photos. a) The CHJ, as seen from the NW. Larger deformation structures are already visible from this distance. See Fig. 1b for location from, and direction in which this photo was taken. b) and c) show typical examples of striated slip planes. d) photograph of a sample cut perpendicular to the slip plane and parallel to the striations. Note the increase in deformation band density towards the slip plane, as well as the bleaching of the sandstone near the slip plane. A picture of the sample is shown in the inset. e) Photomicrograph with crossed polarizers, thin section of the sample shown in e). The deformation bands are clearly cataclastic. Deformation band density increases towards the slip plane at this scale too.**








**Fig. 3: a) Map of the Courthouse Junction outcrop with GPS stations (dots) of the locations were striated slip planes were measured in 35 locations, with 123 measurements. A sketch map of the main structures is also included, but the reader is referred to Eichhubl et al. (2009) and Johansen et al. (2005) for more detailed maps. Also given are the measured striated fault planes, as Angelier lower hemisphere stereographic plot and the Hoepener plot. The small stereograms show faults and striations in the two segments. See Fig. 1 for key. b) Along Segment B and C 15 locations, with 60 measurements are taken (separated in 4 groups; I-IV). See Fig. 1 for key. c) Stereoronets for these groups.**

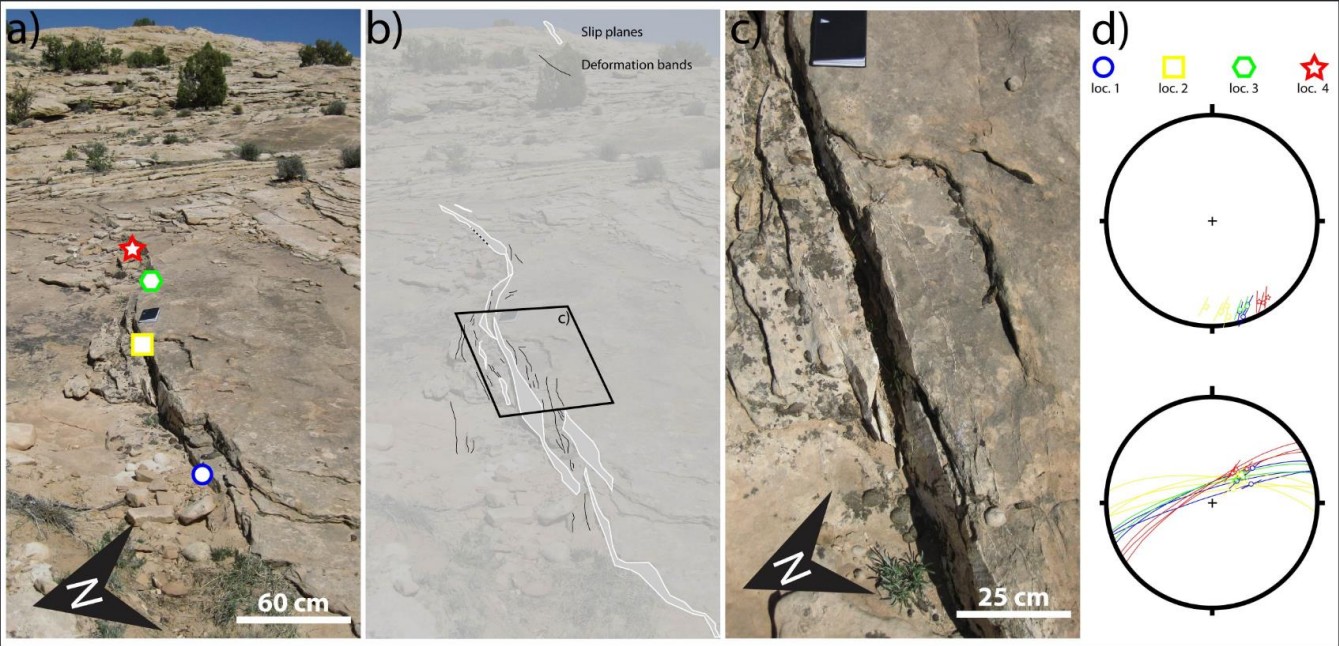

**Fig. 4: A) An undulating slip plane in a bundle of deformation bands. Symbols show measurement locations. For location, see Fig. 3. B) interpreted field image of A). C) detail of A), showing the slip plane in the core of a bundle of deformation bands. D) Stereographic plot of the orientation and slip direction at the four locations, shown in A). Both the pole plot (top) and the great circle plot (bottom) show that these locations exhibit significantly different orientations of the structure, but that the slip directions are similar. The colors and symbols in d) correspond to those in a).**




**Fig. 5: View of the western face of the northern termination of the Courthouse Rock, with visible faults interpreted. The 27 measurements of striated slip planes are recorded at the stations indicated with the black dots. Width of view is about 400 m. The stereonet inset the shows the measured planes and slip directions. See Fig. 1b for location and direction info.**





**Fig. 6: Separated fault orientation –slip direction planes paleostress (DSI) and paleostrain (NDA) results from inside the triangular outcrop of the Courthouse branch point (Fig. 3a). The separation is based on Johansen et al. (2005), but conjugate sets are combined into a single data set here.**

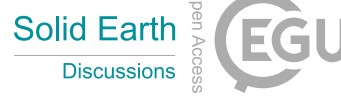

**Fig. 7: Fault orientation –slip direction data and paleostress (DSI) and paleostrain (NDA) results from the in locations I-III along**
**Segments B and C. No paleostress results are shown for location IV, as the density of observations here was too low to obtain a valid result.**



Figure 8: Sketch of the evolution of the Courthouse branch point (CHJ). Broken lines are damage zones in front of normal faults (lines). Blue arrows point in the direction of extension, light blue symbols are inferred. a) The faults are isolated. The Moab fault (Segment A) forms under the influence of the regional NE-SW extension direction (based on fault undulations on segment A and the thick deformation bands set, described by Johannsen et al. (2015). b) When Segment B, initially formed as a parallel fault, starts to rotate and approach Segment A, the segments becomes kinematically linked, and the minimum principal stress rotates at CHJ to NW-SE (Deformation band sets 1&2 and 3&4. c) At the moment Segment A and B connect and become hard-linked or geometrically linked, the minimum principal stress at CHJ rotate back to the regional stress state (deformation band Set 5,6&7).