# Peer review of "Abutting faults: a case study of the evolution of strain at Courthouse branch point, Moab Fault, Utah"

_Solid Earth, 2019_

## Referee Comment (RC1) · Anonymous Referee #1 · 18 Oct 2019

The present study provides new insights into the understanding of local, outcrop-scale stress perturbations within rock volumes encompassing interacting faults. This is a very intriguing research topic, addresses by scientists since the 80s with the germinal work of Angelier and co-authors, which has been recently tackled again by geologists dealing with normal fault linkage and 3D relay ramp geometry. Along this line, the authors submit a research article aimed at assessing the paleo-strain and paleo-stress conditions at the abutting zone of two large faults crosscutting porous sandstones. By combining detailed structural survey of the outcropping fault zones with Numeric Dynamic Analysis (NDA) of the slip vectors and fault planes, the authors calculate the extension directions for not-parallel faults zones exposed at the Courthouse branch

point, Moab Fault, SE Utah (USA). In light of published results on the relative age of the studies structural elements, the results of NDA are discussed in terms of time-evolution of the extension direction during the processes of abutting and linkage of the two normal fault zones. In particular, the authors assess the switch of the main extensional direction from kinematic to geometric linkages (sensu Peacock et al., 2017) of the two fault zones.

The manuscript is well written (although very minor modifications can be made throughout the text, see the Specific Comments below), the aim of the work is clear, the methods robust, and both interpretation and discussion of the original data quite convincing. Overall, it nicely reports the result of a case study that corroborates the current effort on conducting detailed analyses of dimensional and kinematic attributes of interacting fault zones. On this regard, the paper cites the most significant recent articles dealing with this topic. However, a slight improvement of the manuscript can be made by considering the following four points: 1. Authors claim to deal with "thin deformation bands" (cataclastic shear bands), and then interpret their conjugate geometries according to the Navier-Coulomb-Mohr failure criterion. In order to support their interpretation, robust microstructural evidences should be provided. 2. I do not understand the reasons behind the choice of grouping together Sets 5, 6 and 7 made by the authors. According to stereoplots shown in figure 6, the aforementioned failure criterion does not justify this choice. Please explain in the revised text. 3. Since the authors report that cross-cutting relationships among Sets 1&2 and Sets 3&4 were not documented in the field, their relative timing of formation should be better justified. 4. Finally, I recommend to improve the quality of the field structural maps shown in Figure 3 by adding details on attitude and abutting/crosscutting relationships among the various structural elements.

In conclusion, the work done by the authors is fascinating. The topic of the manuscript is interesting, the methods applied are robust and appropriated, and the interpretation quite convincing. However, a minor work on the outcrop and microstructural setting of the investigated DB's, and a better justification to is required before publication. For

this reason, based upon the aforementioned comments, and taking into account the overall quality of the paper, I recommend to accept with minor revisions the submitted manuscript. Specific Comments are reported below.

Specific Comments Abstract: please check for wrong punctuation marks. 1. Introduction: please re-write the first paragraph (too much information, and too many references), and check for wrong punctuation marks. 2. Regional Geology: please re-write lines 56-60 (not clear). 2.1 Outcrop Description: please change the number of this section into 2.1; insert the word "against" after "abuts" (line 67); separate sentences in lines 73 and 74; change the number of this section into 2.2; check for typos and misspells in the second and third paragraphs; delete the words "but these formed in opening Mode I (and are not sheared) in lines 97-98. 3.1 Paleostress and Paleostrain: please check for wrong punctuation marks, and misspells. 3.2 Field Observation: please clarify the meaning of the word "late" (line 163); explain how you know that fractures "results from unloading or weathering close to the surface" (line 163, as well); explain the significance of the word "similar" in lines 167; move the sentence "In agreement with. . .." (lines 175-177) to the Results section. 4.1 Separating the data: please explain the meaning of the word "consecutive" (line 185). 4.2 Paleostrain results: please explain from what the R value is different (line 196); check for wrong punctuation. 4.3 Results in the rest of the study area: please modify the title of this section; add the word "relative" to the age relationships mentioned in line 200; re-write the second paragraph (lines 206-208); move the last sentence (lines 213-214) to the Discussion section. 5. Interpretation and Discussion: please re-write the first sentence (lines 2016-217). 5.1 Deformation during the transition: please explain the reasons behind your interpretation of Segment A being older than Segment B (line 241); please explain why you assessed the formation of Sets 1&2 ahead of the of the fault tip of Segment B; as stated above in the General comments, the whole paragraph reported in lines 251-254 is not clear to me; move the sentences reported in lines 257-259 to the Introduction section.

---

## Referee Comment (RC2) · Fabrizio Balsamo (Referee) · 28 Oct 2019

Dear Editor,

this is my review of the paper SE-2019-137 "Abutting faults: a case study of the evolution of strain at Courthouse branch point, Moab Fault, Utah", by van Gent & Urai, submitted in Solid Earth. The paper is a net study performed in a famous locality in Utah (USA) known as Courthouse branch point, a sector where two normal fault segments of the regional-scale Moab Fault are in contact, and one abut the other. According to this and previous papers, this sector is characterized by outstanding outcrop exposures, which allowed a full structural analysis and fracture mapping at very detailed

scale. Several structural-oriented studies focussed in the past on these spectacular outcrops along the Moab Fault. Such studies are extensively cited by the authors.

The present contribution provides a new structural dataset (n=207) on secondary fault attitude and kinematics that, in combination with cross-cutting relationships and previously published datasets, are analysed and used to constrain the stress/strain evolution in the Courthouse branch point. In particular, seven different fault sets are analyses using numerical dynamic analyses (stress/strain inversions) to obtain the paleo-extension directions through time. The paper is short and well written, and the quality of data is good. Conclusions are sound and supported by the evidences.

There are, however, three weak points that could be strengthened before publication, which mostly concern, in order of importance, the following aspects: (i) amount of structural data presented, (ii) figure quality, (ii) typing errors in the text. Apart from point (i), the two other items can be improved easily. Therefore, I recommend this paper with moderate to major revision.

To facilitate the revision, I list below the major points which should be addressed before publication. Further, I provide in attach a scanned copy with my own modifications and comments on the manuscript.

Kind regards Fabrizio Balsamo

Major points to be addressed (Specific comments) 1) Number of structural data presented In line 165, it is stated that a total of 207 separate fault orientation/slip direction measurements were acquired. So, the stress/strain inversions are based on such data, and the evolutionary model (paleo-extension orientation during faults interaction) is based on this dataset - plus previously published observations by Johannsen et al., 2015. Despite the model in Fig. 8 is supported by the data presented and analysed, I have some general concerns on a structural evolutionary model of a regional fault based on 207 data. Further, in the Methods is explained that, to avoid outliers and measurement errors, each fault surface/slickenline was measured 3 times along their

strike, which makes the fault dataset smaller. If this is the case, it should be stated in the manuscript that this is the only available fault dataset, and that all available faults in the Courthouse branch point were measured and mapped. Moreover, to overcome this issue, a good strategy would be the completion of the structural map in Fig. 3a with the real fault strands (not only straight isolated lines) and the arrows indicating fault kinematics. As it is now, the map in Fig. 3a (Courthouse Junction) contains lines with no kinematic indicators. Finally, I think that some more clear evidence (photos) of cross cutting relationships between the different fault sets should be provided (at least for the 3 groups which constrain the evolutionary model in 3 steps). With the map completed, and more clear evidence of cross cuttings between fault sets, the evolutionary model in Fig. 8 would be better supported.

2) Figure quality Many figures contain letters and symbols which are practically invisible. In particular: Figure 1 – all invisible, either (a) and (b). Please keep in vertical the two maps. Consider that font < 6-7 are invisible in an A4 page. Figure 2 – In Figure 2c the slickenlines are not clearly visible (at least in my printed pdf). Figure 3 – Please in all stereographic projections add the number of data. Also add (a), (b) and (c) for the sub-images. Figure 4 –Increase font size in (b) Figure 5 – Add n data in the stereonet Figure 6 – Add n data in the stereonet (n=XX) and increase significantly the font size in X-Y diagrams. Figure 7 – same as figure 6. Figure 8 – Add (a), (b) and (c) for the three evolutionary steps. Also, indicate segment A and segment B in the fisrt image (isolated faults).

3) Typos and text modifications See the attached PDF with my own corrections.

Please also note the supplement to this comment:
https://www.solid-earth-discuss.net/se-2019-137/se-2019-137-RC2-supplement.pdf

---

## Author Comment (AC1) · 8 Dec 2019

The authors would like to express their thanks to the reviewer for his or her extensive review, and the helpful comments. We have implemented, most, of the changes suggested and address some of his points in detail below. In other cases we clarified our arguments. Points not addressed here section are accepted and will implemented/changed in the final document.

Reviewer's point 1) "Authors claim to deal with "thin deformation bands" (cataclastic shear bands), and then interpret their conjugate geometries according to the Navier-Coulomb-Mohr failure criterion. In order to support their interpretation, robust microstructural evidences should be provided."

We thank the reviewer for raising this point and would like to use this opportunity to explain a little bit better. As described, and published in Johannsen et al. (2005), this outcrop contains 8 sets if "deformation band bundles". In the following we will use the nomenclature of that paper. The bundles of "thick deformation bands" consist of deformation bands (or most likely: disaggregation bands", sensu Fossen et al. (2007 – J Geol Soc L, vol 164 pp 755-769) consisting dominantly of undeformed grains, and 7 sets of different orientated sets of "thin deformation bands" in which the grain size is significantly reduced by cataclasis. The use of "thin" and "thick" is therefor a representation of the relative grainsize (thickness of deformation bands being roughly 3 times the grain size). These deformation bands occur in bundles, of anywhere between 2 to "dozens" of sub-parallel striking, anatomizing deformation bands. Locally within these bundles, highly polished, striated planes are found, which are interpreted by Johannsen et al. (2005) as slip planes, and the striations representing the slip direction. Other authors (Nicholas C. Davatzes et al., 2005; Eichhubl et al., 2009) have interpreted these as (sheared) joints, but the consistent nature of these slip directions, as demonstrated by the NDA analysis shown here, strongly suggest that this is not the case. Also our micrographs (fig. 2) sections show a marked increase in deformation band density towards the slip plane. In fact, the fact that the NDA analysis of the data clustered according to the sets published by Johannsen et al. (2005) into three apparently internally consistent evolutionary steps, shows that these slip planes have not formed under a single stress regime, but are part of bundle they are part of and formed as this bundle is developing. As these slip planes have formed by dominantly frictional processes, the use of the Navier-Coulomb-Mohr failure envelope is justified as first approximation.

Reviewer's point 2) "I do not understand the reasons behind the choice of grouping together Sets 5, 6 and 7 made by the authors. According to stereoplots shown in figure 6, the forementioned failure criterion does not justify this choice. Please explain in the

revised text."

We agree that the stereo plot shown in Figure 6 does not demonstrate these sets to be conjugate, like sets 1 and 2, and sets 3 and 4 are. During the analysis we found that NDA analysis of these three sets of data (5, 6 and 7) were very similar, which led us to combine these sets into one. This point is glossed over in the text, and we will change the text to include this.

Reviewer's point 3) "Since the authors report that crosscutting relationships among Sets 1&2 and Sets 3&4 were not documented in the field, their relative timing of formation should be better justified."

The cross-cutting relationships between the deformation band bundles with in the triangular Courthouse Junction outcrop were indeed not the main focus of this work. Those relationships have been published in detail by Johannsen et al. (2005), and we have only observations that confirm their work. We will clarify this in the revised text. We do however mention that we did not observe any cross-cutting relations ships, between the two sets of slip-plane data points in location I, in the canyon directly south of the outcrop. We will make the text clearer that these relative timings are based only on assumptions of what a stress axes to expect in virgin rock.

Reviewer's point 4) "Finally, I recommend to improve the quality of the field structural maps shown in Figure 3 by adding details on attitude and abutting/crosscutting relationships among the various structural elements"

We agree with this point and will include more structural detail into the map.

Specific Comments (comments not addressed here are already adopted in the final document) • "Please re-write the first paragraph (too much information, and too many references)" We thank the reviewer or this point and will improve the readability of this section. We feel that all references are relevant, but will remove a number "double citations",

• "explain how you know that fractures "results from unloading or weathering close to the surface" (line 163, as well)" This is an assumption not based on data (other than some prior references that indicate "late fracturing"). We will change the text to better reflect this.

• "please explain the reasons behind your interpretation of Segment A being older than Segment B (line 241)" The courthouse junction is assumed to be the result of a breached relay (Foxford et al., 1998, Fossen and Rotevatn, 2016), i.e. two initially parallel faults, where one rotates and grows towards the second until they abut (Rotevatn, 2007, 2009a). In these cases, the parallel faults often are the same age, but the abutting fault needs a pre-existing fault to abut against. Therefore we use the term "older" when describing segment A, compared to segment B but that is only valid at the location of the branch line. However, it is reasonable that further along the fault, there are sections of the faults that are active at the same time. We will adopt the text to use "pre-existing" rather than "older".

• "please explain why you assessed the formation of Sets 1&2 ahead of the of the fault tip of Segment B; " One model of fault growth in lithified rocks is by segment linkage, i.e. the coalesce of smaller, isolated deformation structures into a single through going fault plane (see for example, but not limited to Cartwright et al. (1995 , J Struc Geol, Vol. 17, No. 9, pp. 1319 -1326). One of these early isolated deformation structures in high nett/gross sandstones are deformation bands. We assume therefore that on the current trajectory of Segment B, prior to this segment becoming a fault a range of different deformation structures (fractures, faults, deformation bands) were present. As segment B grew, it incorporated more and more of these isolated structures into its tip line, and essentially freezing those structures that it passed by, as the strain was taken over by the much larger slip plane of segment B. This why we have placed the formations of sets 1&2 in front of the tip of segment B, as they need to exist, prior to being able to be incorporated into the tip line.

---

## Author Comment (AC2) · 8 Dec 2019

We thank the reviewer for his extensive review, and many helpful comments. We have implemented, most, of the changes suggested and address some of his points in detail below. In other cases we clarified our arguments. The reviewer has made his concerns into three mainsections, which we will address below:

Point 1a) "amount of structural data presented" The reviewer is correct in stating that the measured data point represents the vast majority of measurable slip planes in this outcrop. Some slip planes were simply of too low a quality to measure, or were in positions (on cliffs, in cavities) where measurements were not possible. We will adopt

[Figure]

the text to reflect this.

Point 1b) "the completion of the structural map in Fig. 3a with the real fault strands (not only straight isolated lines) and the arrows indicating fault kinematics" We thank the reviewer for this comment. Indeed, the map as it stands now represents more a sketch then a map. The maps published by Johannsen et al (2005), however are excellent and we will redraft the map to incorporate the faults published here.

Point 1c) "more clear evidence (photos) of crosscutting relationships between the different fault sets should be provided (at least for the 3 groups which constrain the evolutionary model in 3 steps). " We have based our work extensively on Johannsen et al. (2005). When working this outcrop we often confirmed their observations, but felt that from a structural mapping/cross cutting study of Johannsen et al (2005) was of such high quality we would not be able to add much. As a result, these cross-cutting relationships were not the focus of our work in this this study. One example of a crosscutting relationship is in attached Figure 1, taken along a scanline of photo's perpendicular to the main fault (scale is in inches). Main fault Segment A is several meters to the left edge of the image and the set of deformation bands of set 3&4 (highlighted in blue) is off-set by the set highlighted in red (from set 5-6-7). We will make it more clear in the text that mapping cross-cutting relations was not the focus of this work, and that we rely on the excellent observations of Johannsen et al. (2005).

2) "Figure quality" All comments here are justified and we will make these changes prior to submitting the final version.

3) "Typos and text modifications" We have implemented all these changes (or slightly modified the text where needed to make the point more clearly). We want to thank the reviewer for including the scanned document, that was very useful!

[Figure]

**Fig. 1.** Cross-cutting relations between two bundles in deformation band set 3&4 (blue) and set 5,6 &7 (red). Scale is in inches

---

## Author Response (AR1)

**Reply to Fabrizio Balsamo**

We thank the reviewer for his extensive review, and many helpful comments. We have implemented, most, of the changes suggested and address some of his points in detail below. In other cases we clarified our arguments. The reviewer has made his concerns into three mainsections, which we will address below:

**(1.1) comments from referees/public**

> 1a) amount of structural data presented

**(1.2) author's response**

The reviewer is correct in stating that the measured data point represents the vast majority of measurable slip planes in this outcrop. Some slip planes were simply of too low a quality to measure, or were in positions (on cliffs, in cavities) where measurements were not possible.

**(1.3) author's changes in manuscript**

We have added lines 192-194.

**(2.1) comments from referees/public**

> 1b) the completion of the structural map in Fig. 3a with the real fault strands (not only straight isolated lines) and the arrows indicating fault kinematics

**(2.2) author's response**

We thank the reviewer for this comment. Indeed, the map as it stands now represents more a sketch then a map. The maps published by Johansen et al (2005), however are excellent (see also next comment) .

**(2.3) author's changes in manuscript**

We have redrafted figure 3, to include the positions , orientations and type (Thin vs thick / set number) from Johansen et al (2005), and also included our own observations.

In addition (not part of this specific comment, but mentioned else were), we included the number of measurements in the steroplots.

**(3.1) comments from referees/public**

> 1c) more clear evidence (photos) of crosscutting relationships between the different fault sets should be provided (at least for the 3 groups which constrain the evolutionary model in 3 steps).

**(3.2) author's response**

We have based our work extensively on Johannsen et al. (2005). When working this outcrop we often confirmed their observations, but felt that from a structural mapping/cross cutting study of Johannsen et al (2005) was of such high quality we would not be able to add much. As a result, these cross-cutting relationships were not the focus of our work in this this study.

One example of a crosscutting relationship is in the image below, along a scanline of photo's perpendicular to the main fault (scale is in inches). Main fault Segment A is several meters to the left edge of the image and the set of deformation bands of set 3&4 (highlighted in blue) is off-set by the set highlighted in red (from set 5-6-7).

[Figure]

[Figure]

**(3.3) author's changes in manuscript**

We have re-written lines 99-108 and added line 108-109 to demonstrate more clearly we rely on Johansen et al (2005) for cross-cuting/age relations.

**(4.1) comments from referees/public**

2) Figure quality

**(4.2) author's response**

All comments here are justified and we will make these changes prior to submitting the final version.

**(4.3) author's changes in manuscript**

We have made the following changes:

Fig 1: improved readability by increasing font size. Moved a) and b) underneath, rather then next to each other. Moved some text from behind the legend box. Added segment names in b). Added * in b).

Fig 2: replaced figure c) by another (higher resolution, more zoomed) image of the same slip plane, as the reviewer mentioned slickenlines were not clearly visible.

Fig. 3: redrafted a) to include the deformation band sets of Johansen et al (2005) and our own observations (with lengths > 5 m).  Added number of observed structures next to the stereograms. B) no changes. C) Added number of observed structures next to the stereograms.

Fig 4. No changes

Fig 5. Changed symbology of measurement stations. Added the number of observed structures in the stereogram

Fig 6. Added number of observed structures next to the stereograms. Increased font size and removed a couple of tickmarks /values along the histogram axes to improve readability.

Fig 7. Added number of observed structures next to the stereograms. Increased font size and removed a couple of tickmarks /values along the histogram axes to improve readability.

Fig 8. Added fault ticks. Added segments names. Increased "crispness" of lines.

**(4.1) comments from referees/public**

3) Typos and text modifications

**(4.2) author's response**

We want to thank the reviewer for including the scanned document, that was very useful!

**(4.3) author's changes in manuscript**

We have implemented all these changes (or slightly modified the text where needed to make the point more clearly. We refer to the attached manuscript with tracked changes for more details.

**Reply to anonymous reviewer**

The authors would like to express their thanks to the reviewer for his or her extensive review, and the helpful comments. We have implemented, most, of the changes suggested and address some of his points in detail below. In other cases we clarified our arguments. Points not addressed here section are accepted and will implemented/changed in the final document.

**(1.1) comments from referees/public**

1) Authors claim to deal with "thin deformation bands" (cataclastic shear bands), and then interpret their conjugate geometries according to the Navier-Coulomb-Mohr failure criterion. In order to support their interpretation, robust microstructural evidences should be provided.

**(1.2) author's response**

We thank the reviewer for raising this point and would like to use this opportunity to explain a little bit better.
As described, and published in Johannsen et al. (2005), this outcrop contains 8 sets if "deformation band bundles". In the following we will use the nomenclature of that paper. The bundles of "thick deformation bands" consist of deformation (or most likely: disaggregation bands", sensu Fossen et al. (2007 – J Geol Soc L, vol 164 pp 755-769) consisting dominantly of undeformed grains, and 7 sets of different orientated sets of "thin deformation bands" in which the grain size is significantly reduced by cataclasis. The use of "thin" and "thick" is taken fom Johansen et al (2005) there for a representation of the relative grainsize (thickness of deformation bands being roughly 3 times the grain size).
These deformation bands occur in bundles, of anywhere between 2 to "dozens" of sub-parallel striking, anatomizing deformation bands. Locally within these bundles, highly polished, striated planes are found, which are interpreted by Johannsen et al. (2005) as slip planes, and the striations representing the slip direction. Other authors (Nicholas C. Davatzes et al., 2005; Eichhubl et al., 2009) have interpreted these as (sheared) joints, but the consistent nature of these slip directions, as demonstrated by the NDA analysis shown here, strongly suggest that this is not the case. Also our cross-sections show a marked increase in deformation band density towards the slip plane (fig. 2). In fact, the fact that the NDA analysis of the data clustered according to the sets published by Johannsen et al. (2005) into three apparently internally consistent evolutionary steps, shows that these slip planes have not formed under a single stress regime, but are part of bundle they are part of and formed as this bundle is developing. As these slip planes have formed by dominantly frictional processes, the use of the Navier-Coulomb-Mohr failure envelope is justified as first approximation.

**(1.3) author's changes in manuscript**

We have re-written section 99-112, and added a reference to Komoroczi (2015) in line 181.

**(2.1) comments from referees/public**

2) I do not understand the reasons behind the choice of grouping together Sets 5, 6 and 7 made by the authors. According to stereoplots shown in figure 6, the forementioned failure criterion does not justify this choice. Please explain in the revised text.

**(2.2) author's response**

We agree that the stereo plot shown in Figure 6 does not demonstrate these sets to be conjugate, like sets 1 and 2, and sets 3 and 4 are. During the analysis we found that NDA analysis of these three sets of data (5, 6 and 7) were very similar, which led us to combine these sets into one. This point is glossed over in the text.

**(2.3) author's changes in manuscript**

We have changed the text in lines 217-218 and elsewhere to reflect this

**(3.1) comments from referees/public**

Since the authors report that crosscutting relationships among Sets 1&2 and Sets 3&4 were not documented in the field, their relative timing of formation should be better justified.

**(3.2) author's response**

The cross-cutting relationships between the deformation band bundles with in the triangular Courthouse Junction outcrop were indeed not the main focus of this work. Those relationships have been published in detail by Johannsen et al. (2005), and we have only observations that confirm their work. We will clarify this in the revised text. We do however mention that we did not observe any cross-cutting relations ships, between the two sets of slip-plane data points in location I (section 4.3), in the canyon directly south of the outcrop.

**(3.3) author's changes in manuscript**

To improve the text we have rewritten lines 99-107 and 250-254.

**(4.1) comments from referees/public**

**4)** Finally, I recommend to improve the quality of the field structural maps shown in Figure 3 by adding details on attitude and abutting/crosscutting relationships among the various structural elements

**(4.2) author's response**

We agree with this point and will include more structural detail into the map.

**(4.3) author's changes in manuscript**

We have redrafted fig. 3a) to include the deformation band sets of Johansen et al (2005) and our own observations (with lengths > 5 m).

Specific Comments (comments not adressed here will be adopted in the final document)

**(5.1) comments from referees/public**

- Please re-write the first paragraph (too much information, and too many references)

**(5.2) author's response**

We thank the reviewer or this point and will improve the readability of this section. We feel that most references are relevant, but will remove a number "double citations" from the same authors.

**(5.3) author's changes in manuscript**

Lines 27-36, removed a number of refernces that were not used elsewhere in the manuscript.

Lines 39-42, removed a reference that was not used elsewhere in the manuscript

**(6.1) comments from referees/public**

- explain how you know that fractures "results from unloading or weathering close to the surface" (line 163, as well)

**(6.2) author's response**

This is an assumption not based on data (other than some prior references that indicate "late fracturing"). We will change the text to better reflect this.

**(6.3) author's changes in manuscript**

We have completely removed the reference to the brown, amorphous iron oxide.

**(7.1) comments from referees/public**

- please explain the reasons behind your interpretation of Segment A being older than Segment B (line 241)

**(7.2) author's response**

The courthouse junction is assumed to be the result of a breached relay (Foxford et al., 1998, Fossen and Rotevatn, 2016), i.e. two initially parallel faults, where one rotates and grows towards the second until they abut (Rotevatn, 2007, 2009a). In these cases, the parallel faults develop during the same period of time but the abutting fault needs a pre-existing fault to abut against. Therefore we use the term "older" when describing segment A, compared to segment B but that is only valid **at** the location of the branch line. However, it is reasonable that further along the fault, there are sections of the faults that are active at the same time.

**(7.3) author's changes in manuscript**

We have adopted the text to make this point clearer

**(7.1) comments from referees/public**

- please explain why you assessed the formation of Sets 1&2 ahead of the of the fault tip of Segment B;

**(7.2) author's response**

One model of fault growth in lithified rocks is by segment linkage, i.e. the coalesce of smaller, isolated deformation structures into a single through going fault plane (see for example, but not limited to Cartwright et al. (1995 , J Struc Geol, Vol. 17, No. 9, pp. 1319 -1326), but also Peacock et al 2017b. One of these early isolated deformation structures in high nett/gross sandstones are deformation bands.

We assume therefore that on the current trajectory of Segment B, prior to this segment becoming a fault a range of different deformation structures (fractures, faults, deformation bands) were present. As segment B grew, it incorporated more and more of these isolated structures into its tip line, and essentially freezing those structures that it passed by, as the strain was taken over by the much larger slip plane of segment B. This why we have placed the formations of sets 1&2 in front of the tip of segment B, as they need to exist, prior to being able to be incorporated into the tip line.

**(7.3) author's changes in manuscript**

We have included a reference in the caption of figure 8 to point to Peacock et al 2017b

---

## Referee Report (RR1)

[referee-annotated manuscript omitted]

---

## Author Response (AR2)

**Author's Response for production version of "Abutting faults: a case study of the evolution of strain at Courthouse branch point, Moab Fault, Utah" (se-2019-137)**

*Heijn van Gent 10/2/2020*

We would like to thank the editor and the reviewers for accepting this paper. In the following we will describe all differences between the version of the manuscript as it was re-submitted in January 2020 and this version, dated February 10th 2020.

We would particularly like to thank the reviewer Fabrizio Balsamo, for taking the time to submit additional notes and proofreading. The document attached below has all track changes marked

- In line 21 we removed the first instance of the word "direction"
- In line 23 we removed the text "in the"
- In line 102 we removed the sentence "In the remainder of this paper we will use this nomenclature."
- In lines 194 – 202 we made the text that was originally bold, normal again
- In line 213-214 we removed the section "of thin deformation band bundle sets" in order to simplify the sentence
- In line 217 we made the text that was originally bold, normal again
- In lines 221-224 we made the text that was originally bold, normal again
- In line 227 we changes the section header to "4.3 Paleostrain results  for the study area outside the CHJ"
- In lines 230-231 we made the text that was originally bold, normal again
- In lines 234-235 we made the text that was originally bold, normal again
- In lines 256-257 & 259-260 we made the text that was originally bold, normal again
- In lines 276, 278 and 280-281 we made the text that was originally bold, normal again
- In lines 293-295 we clarified the text to and removed the word "field"
- In line 353 we added the authors initial.

[revised manuscript text omitted]